# Application of Biosensors for Detection of Pathogenic Food Bacteria: A Review

**DOI:** 10.3390/bios10060058

**Published:** 2020-05-30

**Authors:** Athmar A. Ali, Ammar B. Altemimi, Nawfal Alhelfi, Salam A. Ibrahim

**Affiliations:** 1Department of Food Science, College of Agriculture, University of Basrah, Basrah 61001, Iraq; athmar.ali93@gmail.com (A.A.A.); ammaragr@siu.edu (A.B.A.); nawfalalhelfi@gmail.com (N.A.); 2Food and Nutritional Science Program, North Carolina A & T State University, Greensboro, NC 27411, USA

**Keywords:** biosensors, pathogenic bacteria, bioluminescence, ATP, foodborne

## Abstract

The use of biosensors is considered a novel approach for the rapid detection of foodborne pathogens in food products. Biosensors, which can convert biological, chemical, or biochemical signals into measurable electrical signals, are systems containing a biological detection material combined with a chemical or physical transducer. The objective of this review was to present the effectiveness of various forms of sensing technologies for the detection of foodborne pathogens in food products, as well as the criteria for industrial use of this technology. In this article, the principle components and requirements for an ideal biosensor, types, and their applications in the food industry are summarized. This review also focuses in detail on the application of the most widely used biosensor types in food safety.

## 1. Introduction

Many people around the world become ill each year by consuming food pathogens. These foodborne illnesses are highly correlated to both physical and chemical contamination of foods in addition to the presence of pathogenic microorganisms [1,2]. A number of authors have reported that food contamination caused by microorganisms could be attributed to the natural contamination that occurs in raw materials [3] or the cross-contamination of foods due to different contaminated sources such as air, water, hair, dirt, animal feces, humans, infected wounds, etc. [4].

Microbial pathogens can contaminate foods and cause foodborne diseases [5]. The Centers for Disease Control and Prevention (CDC) in the United States has stated that either foodborne or waterborne pathogens are considered to be the primary causative factors in 76 million cases each year for foodborne illnesses in the United States alone [6]. The percentage of pathogenic bacteria, parasites, and viruses was five million cases, two million cases, and thirty million cases, respectively [7,8].

Multiple conventional tests were applied to detect microbial contaminants in foods, surfaces, utensils, and equipment. These tests included the following: viable cell counting [9], staining [10], carbohydrate fermentation assay, enzyme linked immunosorbent assay [11], polymerase chain reaction [12], ultraviolet detection [13], and fluorescence techniques [14]. Despite the development of many analytical techniques using automated and complex instrumentation for monitoring and detecting the biological contaminants in foods, there are still several drawbacks and limitations to using these traditional approaches [8]. For example, these traditional approaches require large numbers of samples, high skill levels, and are time consuming and costly [15,16]. In addition, most traditional methods require a long time to obtain accurate microbiological results [17]. Consequently, in the past few years, a lot of developed and rapid in situ methods were investigated as an alternative to the existing microbiological approaches. These methods were highly sensitive to count and evaluate food contamination as well as the degree of cleaning and sanitizing of food contact surfaces [18].

Biosensors represent one such innovative method that has been developed to overcome some major problems regarding food sample analysis. Moreover, the use of biosensors to monitor and provide rapid real-time information will be superior compared to traditional microbiological approaches [19]. Adenosine triphosphate (ATP) bioluminescence, a highly effective biosensor, can be used for food process manufacture monitoring such as HACCP (hazard analysis and critical control points) [20,21]. Bioluminescence is the mechanism of light emission from organisms and thereby reflects the chemical conversion of energy into light. The ATP bioluminescence test is since ATP is a significant biological source of energy found in various microbes and thus represents the presence of a living microbe [22].

Biosensor technology was developed to be a useful indicator of bacterial contamination on food and food contact surfaces. In this review, we present the effectiveness of various forms of sensing technologies for the detection of foodborne pathogens in food products, as well as the criteria for industrial use of this technology. This review will also focus in detail on the application of the most widely used biosensor types in food safety.

## 2. Foodborne Pathogens

In recent years, the demand for enhanced food security has gradually increased. As reported in the media and other sources, diseases caused by bacterial contamination represent about 40% in all infections, and the diseases due to foodborne pathogenic have a significant effect on the health of the population as a whole as well as the economy [23].

Foodborne illnesses thus represent an enormous challenge to worldwide health care systems [24]. For example, in the US, about 48 million individuals suffer from foodborne illnesses each year resulting in around 128,000 hospitalizations, 3000 deaths, and $15.6 billion in economic losses [25]. Because human food and water sources can be easily contaminated by a broad spectrum of microbial pathogens, serious illness results if these microbial pathogens or their toxins are consumed [26]. Bacteria, viruses, and parasites are the most prevalent pathogens that cause foodborne diseases [27,28], but fungal foodborne diseases are also identified [29]. Bacteria are the most well-known foodborne pathogen, and cause the greatest number of foodborne illnesses, including the most hospitalizations (63.9%) and deaths (63.7%) [25]. Bacterial contamination can cause repeating intestinal irritation, kidney disease, mental incapacity, receptive joint inflammation, visual impairment, and even death [30]. In addition, foodborne diseases can occur because of toxins produced either from bacteria or fungi, which may survive even after food processing. Foods that are raw, including meat and poultry or vegetables, fruits, eggs, dairy products, and even cooked seafood, can be contaminated with both foodborne pathogens and their toxins [31,32,33]. Examples of foodborne diseases caused by pathogens in the food matrix are shown in Table 1.

## 3. Monitoring of Microorganism Activities in the Food Matrix

A successful microbiological environmental surveillance system can provide early warning of possible microbiological hazards in food items, detect problems, and thereby support comprehensive microbiological safety. Thus, for several decades, the microbiological aspects of food safety have been intensively examined. For example, maintaining food protection has always been a very critical aspect of government policies in some countries. Management systems have been set up to prevent harmful contaminants from being introduced into the food chain [8]. According to the Centers for Disease Control and Prevention (CDC), the influence of microorganisms such as bacteria, viruses, and fungi on human life is worthy of significant attention [22]. The implementation and monitoring of microbial food safety contributes to enhanced productivity, higher wages, sustainable development, and better livelihoods, which is why it has been suggested that policy makers implement appropriate food safety policies in order to enhance global nutrition and improved food security [64].

Microbial food safety is radically different from chemical food safety. Although chemical contaminants and additives usually join the food chain at predetermined levels, microbes may join at any point [65]. Consequently, food regulations everywhere are very straightforward on this level. For instance, the EU General Food Law [66] states: “a high level of protection of human life and health should be assured in the pursuit of community policies”. The microbiological safety of consumer products is also closely linked to the hygienic properties of the manufacturing system. Under these conditions, the implementation of adequate sanitation methods is essential for the protection of the final product. Evaluation of the efficacy of such methods is important for the assurance of these procedures [67]. In fact, all food safety regulations require these inspection activities. Researchers are therefore making considerable efforts to establish rapid and effective methods to meet the requirements of daily investigation and monitoring of food production [67].

The requirement of monitoring contamination in the food chain involves several analytical methods and the use of sophisticated and automated instrumentation that has been recently developed for detection of contaminants in food [68]. However, there are still many drawbacks and limitations to using these traditional approaches [8]. Furthermore, diagnostic tools must be capable of assessing feasibility and flexible enough to identify the pathogen of concern. Table 2 shows a list of some microbiological analysis approaches used to monitor food safety.

## 4. Biosensors

Leland Charles Clark Jr. designed the first biosensor research instrument in 1956 using an electrode to measure the oxygen concentration in blood. After that, scientists from different fields, such as physics, chemistry, and material science, have come together to build more sophisticated, reliable, and mature biosensing devices for applications in the field of medicine [89]. Several approaches using innovative techniques for pathogen enumeration and identification in perishable and semi-perishable foods have been identified in the last few years. In most microbiological research, quantification of bacterial cells is necessary. Therefore, seeking cost-effective techniques with several properties is required, namely high sensitivity, specificity, and fast responses [70,90].

The word biosensor refers to an effective and creative analytical device that has a biological sensing function with a broad variety of applications such as food safety, environmental monitoring, biomedicine, and drug discovery [91]. More specifically, biosensors are widely used in the identification and detection of bacteria and have attracted great interest as one of the most efficient and accurate methods of food analysis and food safety monitoring [92,93,94]. In addition, biosensors typically deliver fast, on-site tracking and thus provide real-time details throughout the production process [95,96]. Biosensors are thus another broad class of bacteria detection method. For example, conductometric measurements provide fast and simple bacterial detection [97].

Because biosensors are analytical devices for the detection of microbial contamination, their function depends on the interaction between biologically active agents, the transducer, and a signal conversion unit [98,99]. Mayer and Baeumner [100] clarified that biosensors typically contain two main components: a target recognition component such as receptors, nucleic acids, or antibodies and a signal transducer that transforms target recognition into physically detectable signals. The internal reflection, fluorescence resonance energy transfer (FRET), chemiluminescence, bioluminescence, and surface plasmon resonance (SPR) have been employed as manufacturing optical transducers in the fabrication of biosensors [8]. In general, biosensors may be divided into three basic groups based on the type of transduction element: optical biosensors, mechanical biosensors, and electrochemical sensors [22]. An example of different components of biosensors used in food analysis is shown in Figure 1. Many compounds, such as bacterial antigens, toxins, microbial contaminated by-products, or spoilage precursors, could be easily detected using biosensors for the rapid analysis of food deterioration and food quality [101].

### 4.1. Types of Biosensors

Biosensors are categorized into various groups depending on their working principles (Figure 2). Examples of biosensors include electrochemical, mechanical, biological, acoustic sensors, surface plasmon resonance (SPR), and optical biosensors. Three of the most important biosensors are discussed below.

#### 4.1.1. Optical Biosensors

Optical biosensor methods characterized by high sensitivity, simple handling, and rapid detection have been used extensively to identify very large numbers of bacteria [102]. Optical biosensors enable visualization of microbial activities in food with the naked eye. The alteration in the transduction surface due to cell connection by means of direct binding or ligand identification assists in active analyte detection. Ivnitski et al. [103] demonstrated that optical biosensors may distinguish microbes in food through either in situ detection in the refractive index or by means of the thickness that develops as bacterial cells attach to receptors on the transducer surface [103]. The optical biological sensor contains a biodegradable polymer by analytical enzymes secreted by microorganisms during the deterioration of the natural product. As the number of bacteria increases, there is increased secretion of enzymes that cause food degradation, which will be visible with the degradation of the polymer [104]. Colorimetric, fluorescence, chemiluminescence, and surface plasmon resonance (SPR) are the principal optical techniques employed [105]. Newly created biosensors for the identification of microbial contamination in food items are shown in Table 3.

Alamer et al. [105] developed an immunoassay with sandwich to diagnose pathogenic bacteria in poultry such as *Salmonella Typhimurium*, *Staphylococcus aureus*, *Salmonella enteritidis,* and *Campylobacter jejuni*. Immobilized lactoferrin on a cotton swab was employed to pick up the bacterial contamination on the surface of the chicken, accompanied by a sandwich immunoassay formulated with a different antibody coupled with colored nano-beads. The form and concentration of the present microorganism defined the color and strength of the cotton swab [105]. Several plant pathogens including the cucumber mosaic virus [106], *Pantoea stewartii* [107], plum pox virus [108], *Prunus necrotic ringspot virus* [109], citrus tristeza virus [110], and potato virus [111] have already been detected using various optical biosensors. SPR biosensors have been used to successfully identify and detect cowpea mosaic virus, tobacco mosaic virus, lettuce mosaic virus, *Fusarium culmorum*, *Phytophthora infestans*, and *Puccinia striiformis* [112].

#### 4.1.2. Electrochemical Biosensors

Electrochemical biosensing techniques are among the most employed platforms for detection of foodborne pathogens [113]. Electrochemical biosensors have been reported to be successful techniques for bacterial detection due to their low cost, accuracy, miniaturization capacity and ability to detect changes directly based on the interaction between the sensor and sample. However, the time required to detect food contamination using electrochemical biosensors has significantly decreased with the advancement of new methods, some of which require as little as 10 min [19]. Electrochemical biosensors are categorized according to the various electrical signals produced by the existence of targets into impedimetric, potentiometric, amperometric, electrochemiluminescent, voltammetric, and conductometric methods [114].

During the last decade, exponential development in electrochemical biosensors has been observed for analysis of food and beverages and to identify genetically modified organisms (GMOs) in food [19]. Chen and colleagues recently established and developed polyaniline- carbon nanotubes (CNTs) as a redox nanoprobe connected to a signal probe to enhance the electrochemical signal for *Mycobacterium tuberculosis* detection [115]. A single-walled carbon nanotube (SWCNT) biosensor was successfully immobilized with a polyclonal antibody to detect *Yersinia enterocolitica* in Kimchi solutions with a low detection of 4 log CFU/mL [116]. The disposable potentiometric paper-based biosensor was designed to detect of *Salmonella Typhimurium*. In the first step, the combination from ethylenedioxythiophene:polystyrene sulfonate was coated on filter paper. Next, antibodies to the target bacteria were covalently attached to filter paper. A linear range of 4.07 log CFU/mL was recorded, with a detection limit of 0.698 log CFU/mL. Less than 5 min was sufficient to perform the analysis and obtain the results [117]. Similarly, Silva and coworkers developed another approach for *Salmonella Typhimurium* detection in apple juice using a potentiometric biosensor conjugating on a gold nanoparticle polymer inclusion membrane, and a detection limit of 6 cells/mL was achieved [118].

#### 4.1.3. Mechanical Biosensors

Mechanical biosensors can measure a mass sensitive sensor surface deflection because the target analytes will be bonded on the functionalized surface [119]. Mechanical biosensors are typically classified into four broad groups according to the sensor-analyte chemical interactions: affinity-based assays, fingerprint assays, separation-based assays, and spectrometric assays [120]. Quartz crystal microbalance (QCM) is a mechanical biosensor that is widely used due to its capacity to track shifts in mass in sub-nanogram amounts. The change in mass using QCM biosensors is recognized by the resonant frequency of quartz crystal, and this technique is commonly used with extreme sensitivity for quantification of the whole cell of microorganisms [121]. Bayramoglu et al. [122] designed A QCM-aptasensor to isolate and rapid detect *Brucella melitensis* in milk and milk products. The aptamer was immobilized on magnetic nanoparticles and the QCM chip for the quantitative detection of *B. melitensis* with high specificity. The QCM biosensor detection limit for determination of *B. melitensis* was 3 log CFU/mL [122].

Lectins were employed and immobilized as a recognition element on the surface of the QCM chip to detect the foodborne pathogen *Campylobacter jejuni*. The limit of detection was 3 log CFU/mL. A modified strategy was utilized to improve the sensitivity of the assay by Masdor et al. [123] who detected *E. Campylobacter jejuni* based on the inclusion of antibody conjugated gold nanoparticles. The limit of detection was enhanced and found to be 2.17 log CFU/mL because the gold nanoparticles exhibited mass amplification effects. Several other studies were successfully employed to develop a novel sensor based on a quartz crystal microbalance with dissipation to detect the most widely spread mycotoxins in red wine called ochratoxin A. The method described here was fast, sensitive, and cost effective, and the analysis time was less than one hour. A limit of detection of 0.16 ng/ml was attained with an excellent linear range between 0.2 and 40 ng/ml [124]. The most advanced mechanical biosensors for the identification of microbial contamination in food items are shown in Table 3.

## 5. Bioluminescence Methods for Detection of Food Contamination

The overall number of microbes is normally calculated using colony plate counts, dilution methods, methods of contact plate and swab, or techniques of membrane filtering. These methods produce repeatable findings that reflect the microbiological contamination. However, the long incubation time of the sample (up to 72 h for bacteria; up to 5 days for fungi) does not allow for rapid correction within one technical process, so for this purpose, tests to estimate the amount of bacteria need to be added quickly [153]. Consequently, Sharpe et al. [154] proposed utilizing the ATP test dependent on bioluminescence. This approach is becoming increasingly common in HACCP program in situ hygiene monitoring. Its principal benefit is the identification of microbial and chemical pollutants within a few minutes.

Recent developments in bio-analytical instruments have allowed for using the capacity of certain enzymes to release photons as a by-product of the enzymes’ reactions. This effect is known as “bioluminescence”, which can be used to identify the cells’ activity. This technique provides results in a short time and is among the latest technologies for rapid microbiological results [155]. Bioluminescence plays an important role in real-time process monitoring due to the emission of bright light by living microorganisms. Some study results also demonstrated that metal ions, heavy metals, phosphorus, naphthalene, genotoxicants and chlorophenols were detected by employing bioluminescence-based biosensors [156]. The bioluminescent organisms in nature are broadly distributed and include a wide remarkably different of species. Among the organisms that emit light are bacteria, dinoflagellates, fungi, fish, insects, shrimp, and squid. The enzyme luciferase is responsible for catalyzing the bioluminescence reactions that occur in these organisms, and in certain instances the substrates are referred to as luciferins. Bioluminescence is very effective when used for fast spot tracking because tests are obtained in less than 15 minutes [157]. This procedure has been used on several food items including fresh and pasteurized dairy products [158], meat and poultry products [159], beer [160], and fruit products [161].

Sanitizing programs and hazard analysis and critical control point (HACCP) programs can be achieved in the food processing industry by using the common bioluminescence method of adenosine triphosphate (ATP). Bioluminescence assays and the identification of bacterial adenosine triphosphate (ATP) are strong predictors of the occurrence of food contamination in meat, poultry and dairy products and the cross-contamination of surfaces [162]. All living organisms use ATP to store energy. ATP acts as a chemical energy storage unit for free energy that is emitted through catabolism and thereafter used for anabolic processes [163]. The amount of ATP specifically reflects the presence of metabolic cells and can be used to count viable living cells in samples. This is because there is a linear association between the total number of available ATP molecules and the total number of colony-forming units, especially in bacteria and yeast [164].

The relationship between microbial biomass and intercellular ATP can be used to quantify the total number of microorganisms in food items. Recent studies have shown that the amount of ATP present in a cell differs based on the species and growth states of microorganisms. For instance, the extracellular ATP present in *Acinetobacter junii* and *Pseudomonas aeruginosa* at an incubation time of 6 h was 255.2 ± 56.8 nM/OD and 25.5 ± 1.1 nM/OD, respectively [165]. Xu et al. [88] developed the traditional ATP fluorescence detection system by using a rapid detection system based on a nanoprobe and graphite electrode coupled with ATP bioluminescence technology for *Escherichia coli* detection in food. With this new approach, the researchers were not only able to use the probe to capture and enrich *Escherichia coli* via an antibody–antigen reaction, they were also able to enrich ATP using an electric field generated by the graphene transparent electrode (GTE) in order to improve the accuracy of the system. This method resulted in the successful generation of a linear correlation coefficient of up to 0.972 compared to other traditional methods and satisfied the design criteria. The analysis was obtained within 20 min. The system was able to detect the total bacteria count in the range of 2–6 log CFU/mL, and its precision has a CV of 4.2%, indicating good reliability and repeatability [88].

Moreover, Fan and colleagues confirmed the possibility of developing a bioluminescence-based ATP assay using antibacterial peptide-coated magnetic spheres to distinguish Gram-positive G^+^ bacteria from Gram-negative G^−^ bacteria. The authors obviously found the conventional bioluminescence-based ATP cannot distinguish G^+^ bacteria from G^−^ ones since ATP can be released from both bacterial cells. The results exhibited a linear range for G^+^ bacteria between 3.36 and 7.07 log CFU/mL, and the limit of detection was 2.34 log CFU/mL within 33 min [166].

## 6. Principle of Bioluminescence Based-ATP Determination

Adenosine triphosphate is the main activated energy carrier of all living cells in nature, including bacteria, mold, yeast, and algae [167]. ATP levels can also be used as a criterion for microbial activity measurement. ATP bioluminescence is based on a biochemical reaction catalyzed by the enzyme [168]. The reaction is catalyzed by the luciferase enzyme conversion of luciferin to oxyluciferin in the presence of oxygen (O_2_) and magnesium cation (Mg^++^), and ATP adenosine triphosphate is converted to adenosine monophosphate (AMP) with the emission of light [169]. The intensity of light in the luminescence reaction is expressed in relative light units (RLU). The reaction between ATP and luciferin and luciferase complex is described according to the following equation:(1)$$\mathrm{Luciferin}\;+\mathrm{ATP}+\;O_{2}\;\;\;\overset{luciferase,\;Mg\dot{+2}}{\to }\;\;\mathrm{Oxyluciferin}+\mathrm{AMP}+prodcuts+\mathrm{light}\;$$

This light output from the breakdown of cellular ATP by the bioluminescence reaction can be measured using sensitive photons of light meters in an instrument called a luminometer. The greater the amount of ATP will present, the higher amount of light produced by the APP assay test; consequently, the greater the RLU level produced. ATP bioluminescence has often been used for the investigation of microbial contamination of food contact surfaces and for measuring the efficiency of cleaning procedures. It is a simple and rapid method that provides results within minutes compared to conventional methods, which typically take 24–48 h. Libudzisz and Kowal and [170] stated that on the bacterial cell possesses approximately 1 ATP femtogram. Based on the species, physiological status or metabolic function of microorganisms, the concentration will vary from 0.1 to 5.5 fg/cell. Luo et al. [171] claimed that the average concentration of ATP in a cell is approximately 0.47 Cell fg. To determine the number of microbes in each sample, it is presumed that 1 pg of ATP is equal to 1000 bacterial cells. Table 4 below shows the content of ATP (fg/cell) in some bacterial, mold, and yeast cells.

## 7. Applications of Bioluminescence Based ATP in the Food Industry

### 7.1. Hygiene Monitor

The efficacy of ATP-based bioluminescent assays is enhanced due to their ability to provide rapid results that indicate the existence or absence of certain biological contaminants in real time [176]. ATP bioluminesce assays are widely used in the food industry for estimating the cross-contamination of surfaces and products through swabbing. This type of application enables results within 5 min that are just as accurate as those obtained using traditional techniques. The levels of overall surface contamination can be indicated successfully because ATP from all microbial sources will be detected [177]. The time of bacterial viability on certain kitchen surfaces ranges between four and 24 h. Therefore, during food preparation it is necessary to design appropriate hygienic protocols such as proper washing and disinfection to control and avoid microbial risks. The ATP test thus helps to quickly verify that surfaces are clean and properly disinfected. In addition, this method does not pose a threat to humans [178]. However, because raw materials of plant or animal origin increase ATPs, the test results can be overstated. About cleanliness and hygiene, it is not known yet whether microorganisms or traces of biological content are found throughout the work and the production equipment by measuring only the ATP [179]. In this case, the values are usually dependent on the relative light units (RLU) rather than the concentration of ATP collected. The findings are correlated with the previously defined baseline levels for the industry and the individual measurement points. Low RLU rates would mean that the measurement point is safe and clear of chemical and microbiological contaminants, while high RLU levels would be indicative of points of contamination [179]. In a study conducted by Rodrigues et al. [180], the relationship between the values of ATP-bioluminescence and the extent of microbial contamination was estimated according to traditional methods in order to evaluate the cleanliness of the cutting surfaces in the poultry slaughterhouse [180]. Their findings confirmed that that there was a linear relationship between the microbial content using conventional methods and the bioluminescent ATP approach. Using the bioluminescent ATP detection system, extremely low contamination rates can be identified in seconds, enabling a rapid assessment of the surface hygiene [180].

Despite rapid hygiene monitoring using ATP tests, recent studies by Bakke and Suzuki [181] who reported that ATP could be hydrolyzed by heat treatment, acidic factors or alkaline conditions to ADP and AMP. Consequently, the values of collected RLU will not be accurate. Bakke and Suzuki [181] have developed a novel hygiene monitoring based on the detection of total adenylate (A3) in a wide variety of foods such as fermented foods, dairy, vegetables, meat, nuts, seafood, and fruits. After thorough washing with detergent and rinsing the stainless steel, the amount of collected RLU of A3 was 200. In contrast, less than 200 RLU was seen on a traditional ATP system. In conclusion, the A3 assay seems to be a successful approach and more sensitive for detecting adenylates from food residues that are not identified by traditional ATP assays [181].

### 7.2. Milk and Dairy Products

The shelf life of milk depends on its initial microbial load, the form and distribution of microbes, and how well such microbes grow under different storage conditions. Conventional qualitative and quantitative methods were applied in microbiological analysis of food to detect microbial contamination using a selective media, non-selective media and biochemical screening [182]. These approaches are time-consuming and require additional confirmation and interpretation by qualified technicians, which can take several days. Therefore, an alternate, fast, efficient, and lower cost method for real-time identification of milk spoilage is warranted [183]. Recently, the bioluminescence-based ATP technique has been developed to monitor the presence of microorganisms and can easily be applied to determine both somatic cell counts (SCC) and microbial counts for controlling raw milk production quality [178,184]. After treatment with a non-ionic detergent, an indication of the somatic cell concentration in milk can be obtained from the ATP concentration level. This result can be considered as an indicator for infection with mastitis [178]. Indeed, Moore et al. [185] reported that ATP bioluminescence procedures were performed in 5–10 min to detect as few as 4 log CFU/mL of milk bacteria which undoubtedly resulted in faster and better-informed decisions regarding the status of incoming milk tankers the milk processing industries.

Other studies have examined the use of the bioluminescence -based ATP technique compared to total bacterial count (TBC) cultivation for rapid microbial identification to monitor ultra-high temperature (UHT) milk quality [186]. ATP bioluminescence was suitable for detecting very low concentrations of microbial content compared to results for conventional total bacterial counts, and the analysis time was only 20 min. Similarly, Lomakina and others used a bioluminescence ATP assay to ascertain the quality of milk within 20 min with a detection limit of approximately 1.11 log CFU/mL [168].

### 7.3. Meat and Meat Products

Meat and meat products can be used effectively as rich media for growing several microflora (bacteria, yeasts, and molds), some of which are pathogens [187]. The ATP bioluminescence method was used to monitor the microbial content of meat. The study reported that there was a significant correlation between the content of ATP and total bacteria counts of vacuum-packed cooked cured meat products, and a detection limit of 5–6 log CFU/g was sufficient for screening purposes [188]. Similarly, Siragusa and colleagues established a quick ATP assay to quantify total bacteria counts in beef and pork carcasses in commercial food industries and to compare findings with the standard method of viable plate counts using correlation analysis [189]. The results of this research showed that the correlation coefficient between the conventional microbiological assay and the ATP method was 0.91 for beef and 0.93 for pork carcass samples. The ATP test applied linearly to microbial contamination rates > log 2.0 aerobic CFU/cm^2^ in carcasses of beef and > log 3.2 aerobic CFU/cm^2^ in carcasses of pork. The ATP test including sampling took approximately 5 min [190].

However, one concern with this approach is the presence of ATP in meat and in all living cells. Therefore, ATP must be destroyed before an ATP bioluminescence method can be performed to measure only the microbial ATP produced [190,191]. Hence, Cheng et al. [190] conducted an experiment to combine an ATP bioluminescence assay with functional magnetic nanoparticles (FMNPs) for rapid isolation and detection of *Escherichia coli* from artificially contaminated ground beef. To release the target bacterial ATP in the presence of luciferin–luciferase mechanism, immune particles were used to precisely capture and separate the bacteria to generate the luminescence signal. *E. coli* bacteria can be calculated with a detection limit of 1.30 log CFU/mL in the range of 1.30–6.30 log CFU/mL. The whole process used to identify *E. coli* took approximately 1 h. The range of identification and assay time obtained in this study has been shown to be superior to that of other techniques [190].

### 7.4. Fish and Fish Products

For more than 50 years, ATP and associated compounds have been used for the quality evaluation of fish and shellfish [192]. Bioluminescence is the production and release of light by a living entity and exists commonly in aquatic vertebrates and invertebrates. Shim et al. [193] measured the ATP content in the muscle of olive flounder (*Paralichthys olivaceus*) by calculating the intensity of light released using luciferase provided by American fireflies. The findings of bioluminescence were nearly equal to high-performance liquid chromatography (HPLC). Indeed, the results of the study showed a high correlation of r^2^ = 0.98 between luminometer-measured RLU and HPLC-based ATP content. Tanaka et al. [194] have established a bioluminescence system for the identification of AMP in the Atlantic bonito (*Sarda sarda*). Polyphosphate (polyP)-AMP phosphotransferase (PPT) and adenylate kinase (ADK) were utilized from the *Acinetobacter johnsonii* strain conjugated with firefly luciferase. With this approach, the researchers were able to identify high-sensitivity AMP in food residues [194]. Regarding the evaluation of different microbiological methods, Gram [195] found that the correlation between bacterial ATP levels and plate counts was 0.97–0.99 for four fish species. During storage trials, the ratio of bacterial ATP to total count bacteria remained constant and did not vary significantly among fish species [195]. As the amount of ATP per cell varies based on nutritional conditions, stress, etc., it is advised that a standard curve for each specific product be generated [196].

Other experiments conducted by Miettinen et al. [197] reported the presence of *Listeria* in 28 fish processing factories and the extent of surface contamination utilizing specific approaches such as total aerobic heterotrophic and enterobacteria, yeast and mold tests and ATP levels. ATP tests and the total bacteria contact agar slide methods were negatively associated (r = 0.21). However, for both methods, 68 percent of the samples were rated as decent to fair or unacceptable. The microbiological limit of 1 RLU using an ATP assay was exceeded in 43.3% of the samples. The results of this study confirmed that the ATP system recognized 18.1% of the samples that were considered contaminated per the results of the contact agar slide process, and 13.6% of the samples allowed by the contact agar slide system were rejected by the ATP process [197].

## 8. Advantages and Disadvantages of ATP Bioluminescence

ATP bioluminescence provides a better image of the reaction to the contaminant by presenting physiologically relevant data. Bioluminescence is fast and simple to calculate, resulting in the in-situ detection of a wide range of microorganisms. The bioluminescent sensors of whole cells have benefits over conventional approaches by being faster, more cost effective, easy to carry out and less labor-intensive [198]. While not an alternative to traditional approaches, an ATP-bioluminescence-assay can also be a valuable tool for determining the efficacy of environmental cleanliness procedures even with very low microbial counts [199]. Moreover, bioluminescent techniques often possess several benefits compared to fluorometric techniques mainly because no wavelength of excitation is required for the representation of light. In addition, unlike the fluorescent labeling of bacterial species, there is a total energy reliance on the emission of bioluminescents, which enables the capability to distinguish between living and dead cells. Consequently, bioluminescence is a highly valuable instrument for regulating in situ microbial deterioration and is thus a desirable tool for hygiene efficacy [200].

Luminescent approaches often pose some general disadvantages. The most significant disadvantage is the quenching of released light, which negatively influences measurements. The sum of light determined photometrically may be greatly decreased by molecules from the biological samples. However, the biological samples produce certain luminescent non-microbial substances that increase the intensity of the measured light. Bacterial bioluminescent assays are thus capable of being a liability in the food microbiology industry. For example, the results of bacterial bioluminescent assays can be false negatives or false positives by using phage or plasmid host ranges that are either too specific or too extensive [177]. Another disadvantage of bacterial bioluminescent assays is their unreliability about efficiently identifying gram-negative bacteria due to the incomplete lysis of the cells [201].

## 9. Conclusions and Future Directions

Developing biosensors with the necessary properties for reliable and effective use in routine applications is challenging. Despite the great effort spent on the development of various types of biosensors over the past few years, only a few for bacterial detection are commercially available or are approaching commercialization. Requirements for ideal sensors include the specificity to distinguish the target bacteria in a complex food product, sensitivity to detect bacteria directly, and the ability to provide real-time results within a reasonable time. Detection of pathogen or toxic chemicals in food matrix is not a simple and rapid approach. Indeed, it requires additional preparation steps before detection. This includes sample preparation and harvesting the target microbial cells or chemical. The development of any rapid biosensors for detection of pathogens also relies on the type of food products and the nutrients present in these products, such as fat, proteins, and fibers. Hence, there might be a need to develop a specific sensor for each food product or specific analytical tools and sampling methods.

This review highlights potentially reliable biosensor methods to expand research in this area and to address the need for the development of more economical and cost-effective methods. In addition, there is a need to develop a portable bioluminescence-based ATP unit that can be utilized on farms to detect pathogens on the surface of fresh produce. Moreover, such biosensors should provide reliable results in addition to being easy and simple to use without the need for consumer training.

## Figures and Tables

**Figure 1 biosensors-10-00058-f001:**
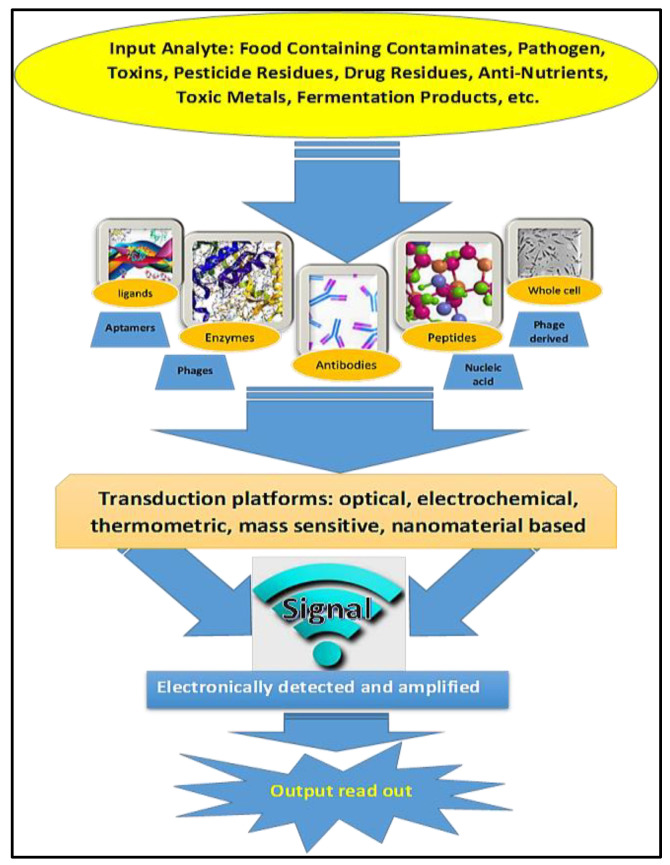
Diagram showing the different components of a biosensor used in food analysis.

**Figure 2 biosensors-10-00058-f002:**
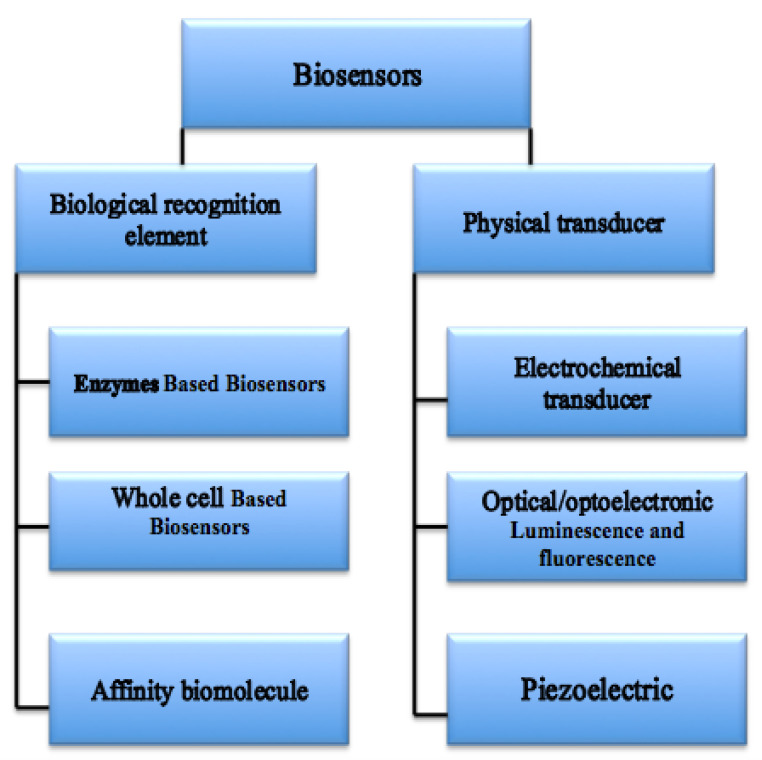
Schematic representation of various combinations of physical and biological elements of biosensors.

**Table 1 biosensors-10-00058-t001:** Examples of Foodborne Diseases Caused by Microorganisms in the Food Matrix.

Pathogenic Sources	Food Matrix	Symptoms and Illnesses	References
*Staphylococcus aureus*	Unpasteurized Milk and Cheese Products	Food Poisoning	Khare et al. [34] Mostafa et al. [35]
*Bacillus cereus*	Dairy Products, Dry Foods, Rice, Egg Products	Diarrhea, Vomiting	Grutsch et al. [36] Griffiths and Schraft [37]
*E. coli O157:H7*	Meat Products and Milk	Diarrheal Diseases and Producing of Shiga Toxins	Xu et al. [38] Kramarenko et al. [39]
*Vibrio parahaemolyticus*	Seafood	Diarrhea	Letchumanan et al. [40] Jiang et al. [41]
*E. coli* O26	Ground Beef	Stomach Cramps, Bloody Diarrhea, Vomiting and High Fever	Ma et al. [42] Amagliani et al. [43]
*Salmonella enteritidis*	Meats, Eggs, Fruits, Vegetables	Vomiting, Diarrhea, Cramps, Fever	Sharma [44] Paramithiotis et al. [45]
*Vibrio parahaemolyticus* *Vibrio cholerae*	Freshwater Fish and Shellfish	Severe Diarrhea, Cholera	Li et al. [46] Baron et al. [47]
*Klebsiella pneumoniae*	Fresh Fruits and Vegetables	Pneumonia	Mesbah Zekar et al. [48] Ghafur et al. [49]
*Campylobacter jejuni*	Meat, Poultry	Postinfectious Reactive Arthritis	Riley [50] Skarp et al. [51]
*Clostridium perfringens*	Poultry Meat	Human Gastrointestinal Diseases	Hamad et al. [52] Rouger et al. [53]
*Clostridium botulinum*	Uncooked Food, Canned Foods	Botulism	Aston and Beeching [54] Yadav et al. [55]
*Listeria monocytogenes*	Lentil Salad	Gastroenteritis and Invasive Infection	Drali et al. [56] Vojkovska et al. [57]
*Shigella* sp.	Poor Water Supply	Watery Diarrhea Mixed with Blood and Mucous	Nisa et al. [58] Shafqat et al. [59]
hepatitis E virus	Rabbit Meat	Liver Disease	Bigoraj et al. [60] Kaiser et al. [61]
*Salmonella*	Fresh Vegetables	Gastroenteritis	Yang et al. [62] Saw et al. [63]

**Table 2 biosensors-10-00058-t002:** Examples of Microbiological Analysis Approaches for Monitoring Food Safety.

Microbiological Approaches	Detection Limit (Log CFU/mL)	Time Consumed	References
Viable Cell Counting	Unlimited	days	Rajapaksha et al. [9] González-Ferrero et al. [69]
Microscopy	Unlimited	min	Sakamoto et al. [70] Mobed et al. [71]
Absorbance	8–9	Immediate	Hazan et al. [72] Ikonen et al. [73]
Enzyme Linked Immunosorbence	2.83–3	3 h	Shen et al. [74] Preechakasedkit et al. [75]
Staining with Fluorescence Dyes	3–4	26 min	Guo et al. [76] Annenkov et al. [77]
Start Growth Time	1.60–2.60	h	Hazan et al. [72]
Flow Cytometry	4–8	h	Ou et al. [78] Adan et al. [79]
Methylene Blue Dye Reduction Test	7	h	Bapat et al. [80] Pawar et al. [81]
Isothermal Microcalorimeters	>2	5–7 h	Fricke et al. [82] Broga et al. [83]
Laser-Induced Breakdown Spectroscopy (LIBS)	1	3 min	Multari et al. [84] Moncayo et al. [85]
Fourier Transform Infrared (FT-IR) Spectroscopy	5.3	60 s	Ellis et al. [86] Johler et al. [87]
Nanoprobe-ATP	2–6	20 min	Xu et al. [88]

**Table 3 biosensors-10-00058-t003:** Newly Created Biosensors for the Identification of Various Contaminants in Food Items.

Type of Sensor	Contaminant	Food Items	Detection Limit	Consuming Times	Reference
**Optical Biosensor**					
Chemiluminescence	*Listeria monocytogenes*	Milk	1.1 log CFU/mL	40 min	Shang et al. [125]
Colorimetric	*Cronobacter* *sakazakii*	Powdered Infant	3.85 log CFU/mL	30 min	Kim et al. [126] Shukla et al. [127]
localized Surface Plasmon Resonance (LSPR)	*Salmonella typhimurium*	Pork Meat	4 log CFU/mL	30–35 min	Oh et al. [128]
Interferometric	*Escherichia coli*	Buffer	0.34 log CFU/mL	2 h	Zaraee et al. [129] Janik [130]
Surface Plasmon Resonance (SPR)	*Pseudomonas*	Water	7.09 log CFU/mL	25 min	Mudgal et al. [131] Zhang et al. [132]
**Mechanical Biosensor**					
Multi-Channel Series Piezoelectric Guartz Crystal (MSPQC)	*Mycobacterium tuberculosis*	Buffer	1 log CFU/mL	1 day	Ren et al. [133] He et al. [134]
Quartz Crystal Microbalance (QCM)	Salmonella	Milk	2 log CFU/mL	10 min	Ozalp et al. [135] Farka et al. [136]
QCM	*Campylobacter jejuni*	Poultry	1.30 log CFU/mL	30 min	Wang et al. [137] Masdor et al. [138]
QCM	*Staphylococcus aureus*	Buffer	7.41 log CFU/mL	1 day	Pohanka [139] Noi et al. [140]
**Electrochemical**					
Potentiometric	*Staphylococcus aureus*	Pig skin	2.90 log CFU/mL	2 min	Zelada-Guillén et al. [141] Arora et al. [142]
Impedimetric	*Salmonella Typhimurium*	Apple Juice	0.47 log CFU/mL	45 min	Sheikhzadeh et al. [143] Bagheryan et al. [144]
Amperometric	*Streptococcus agalactiae*	Fish	1–7 log CFU/mL	90 min	Vásquez et al. [145] Arachchillaya [146]
**Electrochemical Chemiluminescence (ELC) Biosensors**					
Aptamer-Based ECL Sensors	*Escherichia coli*	Luria–Bertani Broth	0.17 CFU/mL	40 min	Hao et al. [147]
ECL Immunosensor	*Vibrio parahaemolyticus*	Seafood	0.69 log CFU/mL	1 h	Sha et al. [148]
Paper-Based Bipolar electrode ECL	*Listeria monocytogenes*	Buffer	10 copies/μL	10 s	Liu and Zhou [149]
**Photoelectrochemical Biosensors**					
label-Free Photoelectrochemical Aptasensor	Bisphenol	Milk	0.5 nM	90 s	Qiao et al. [150]
Tungsten Disulfide (WS2) Nanosheet-Based Photoelectrochemical	Chloramphenicol	Milk Powder	3.6 pM	105 min	Zhou et al. [151]
Visible-Light Photoelectrochemical Aptasensing	Sulfadimethoxine	Milk	0.55 nM	50 s	Okoth et al. [152]

**Table 4 biosensors-10-00058-t004:** The Content of ATP (fg/cell) in Some Bacterial, mold and Yeast Cells.

Microorganisms	ATP (fg/Cell)	References
*Campylobacter jejuni*	1.7	Ng et al. [172]
Yeast	100	Miller and Galston [173]
*Lactobacillus* sp.	2.0–2.2	Libudzisz and Kowal [170]
*Pseudomonas fluorescens*	0.6	Pistelok et al. [174]
*Escherichia coli*	1	Libudzisz and Kowal [170]
Bacteria Mixture	1	Miller and Galston [173]
*Lactobacillus acidophilus*	0.33	Nelson [175]
*Campylobacter coli*	2.1	Ng et al. [172]

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
