# Peer review of "Application of Biosensors for Detection of Pathogenic Food Bacteria: A Review"

_biosensors, 2020, doi:10.3390/bios10060058_

Round 1

Reviewer 1 Report

The present review entitled "Application of Biosensors for Detection of Pathogenic Food Bacteria: a review" is under consideration for publication into Biosensors. The manuscript presents an overview on the potential application of biosensors to detect infective bacteria into food samples, with emphasis on the different biosensors natures, principles, towards food categories. The present manuscript could help researchers in approaching biosensors fields, elucitading also advantages and drawbacks of such systems in this field. All the paragraphs are clear, fluent and grammatically correct. So that, I recommend the acceptance of the manuscript.

Good luck to the authors!

Author Response

The authors would like to thank the reviewer 1 for the valuable comments.

Reviewer 2 Report

This review is for the biosensor for the detection pathogen in food.

As author may know, it is really hard to detect pathogen or toxic chemicals in food by using molecular (including affinity) tools. I think main hurdles to detect them in food are 1.matrix diversity and complexity, 2. sample size, and 3. target variety.

Therefore,

I recommend followings

1. It would be better for author to add how difficult to isolate (concentrate and purify) the target (pathogen and so on) from food samples in introduction section or new section. In there, sample preparation techniques could be also mentioned.

2. Classification of target regarding as food type (drinking water, dairy products, vegetables, meat and so on) would be required.

3. I think author can mention about lab chip or micro total analytical system (sample preparation and sensing part integrated) in the food fields.

Author Response

Greetings:

We would like to sincerely appreciate your valuable suggestions on our manuscript (Manuscript ID: biosensors-802959). Please note that we revised the manuscript taking into consideration all your valuable comments.  I have attached 2 documents to review.   The revised manuscript and our comments to all reviewers.  

Reviewer 3 Report

The manuscript summarized various forms of sensing technologies for the detection of foodborne pathogens in food products as well as the criteria for industrial use of this technology. In particular, the ATP-based bioluminescent assays was discussed in detail on the principle components and application in food safety. Specific comments,

1 In Table 3: the ‘Single-walled carbon nanotube (SWCNT)’ should be moved out from electrochemical classification. The ‘Chemiluminescence biosensors’ should be included in ‘Optical biosensor’ section, and ’electrochemical chemiluminescence biosensors’ and ‘photoelectrochemcal biosensors’ should be included in the Table 3.

2 Currently used biomolecules (biomolecular probes) for recognition of pathogenic food bacteria should be added in the Table 1.

3 The assaying times of mentioned biosensors should be added in Table 3.

Author Response

Reviewer 3

In Table 3: the ‘Single-walled carbon nanotube (SWCNT)’ should be moved out from electrochemical classification. The ‘Chemiluminescence biosensors’ should be included in ‘Optical biosensor’ section, and ’electrochemical chemiluminescence biosensors’ and ‘photoelectrochemcal biosensors’ should be included in the Table 3.

Single-walled carbon nanotube (SWCNT) has been removed. The Chemiluminescence biosensors had been added in the Optical biosensor section. Both electrochemical chemiluminescence biosensors and photoelectrochemcal biosensors have been added in the Table 3.

2-Currently used biomolecules (biomolecular probes) for recognition of pathogenic food bacteria should be added in the Table 1.

I think that the reviewer meant to add the biomolecular probes for recognition of pathogenic food bacteria in the Table 2 not table 1. The biomolecular probes has been added as requested

3-The assaying times of mentioned biosensors should be added in Table 3.

The consuming time of mentioned biosensors has been added in the table 3

Round 2

Reviewer 2 Report

I understood author's point. Regarding as sensor, it is good enough to be published to here. 

Reviewer 3 Report

The revised manuscript could be published as it.